# A new mode of pancreatic islet innervation revealed by live imaging in zebrafish

Yu Hsuan Carol Yang[1]*, Koichi Kawakami[2,3], Didier YR Stainier[1]*

[1]Department of Developmental Genetics, Max Planck Institute for Heart and Lung Research, Bad Nauheim, Germany; [2]Division of Molecular and Developmental Biology, National Institute of Genetics, Mishima, Japan; [3]Department of Genetics, SOKENDAI (The Graduate University for Advanced Studies), Mishima, Japan

**Abstract** Pancreatic islets are innervated by autonomic and sensory nerves that influence their function. Analyzing the innervation process should provide insight into the nerve-endocrine interactions and their roles in development and disease. Here, using in vivo time-lapse imaging and genetic analyses in zebrafish, we determined the events leading to islet innervation. Comparable neural density in the absence of vasculature indicates that it is dispensable for early pancreatic innervation. Neural crest cells are in close contact with endocrine cells early in development. We find these cells give rise to neurons that extend axons toward the islet as they surprisingly migrate away. Specific ablation of these neurons partly prevents other neurons from migrating away from the islet resulting in diminished innervation. Thus, our studies establish the zebrafish as a model to interrogate mechanisms of organ innervation, and reveal a novel mode of innervation whereby neurons establish connections with their targets before migrating away.
DOI: https://doi.org/10.7554/eLife.34519.001

*For correspondence:
Carol.Yang@mpi-bn.mpg.de
(YHCY);
Didier.Stainier@mpi-bn.mpg.de
(DYRS)

## Introduction

Pancreatic islets are innervated by sympathetic, parasympathetic and sensory nerves (*Taborsky et al., 1998*; *Havel and Ahren, 1997*; *Ahrén, 2000*; *Gilliam et al., 2007*). Studies have highlighted the role of autonomic innervation on pancreas development and endocrine hormone secretion (*Ahrén, 2000*; *Borden et al., 2013*). Whether innervation directly modulates glucose homeostasis and contributes to diabetes progression remains controversial, despite effects of sympathetic and parasympathetic nerves on pancreatic islet hormone secretion (*Ahrén, 2000*; *Gautam et al., 2006*) and the role of sensory fibers on autoimmunity (*Tsui et al., 2007*). The debate stems in part from a lack of detailed understanding of several processes including the innervation process during development, the maintenance of innervation in adult stages, and the communication between nerves and islet cells. Although differences in autonomic innervation pattern in the endocrine pancreas have been observed between rodents and humans (*Rodriguez-Diaz et al., 2011*, *2012*), the influence of the autonomic nervous system on pancreatic hormone secretion is conserved across species (*Havel and Ahren, 1997*; *Ahrén, 2000*; *Gilliam et al., 2007*; *Borden et al., 2013*; *Gautam et al., 2006*). More recently, this difference in pattern has been challenged by immunostaining of thick tissue sections and whole mounts (*Hsueh et al., 2017*; *Tang et al., 2018*, *2014*). Given the age-related disparity between rodent and human samples in studies comparing innervation architecture, the dynamic regulation of pancreatic innervation in response to age and environmental influences could be contributing to the observed species differences. Indeed, higher innervation density is observed in human fetal pancreas in comparison to adult (*Proshchina et al., 2014*), and loss of innervation is observed in humans with Type one diabetes (*Mundinger et al., 2016*), as it is

in several rodent models of autoimmune diabetes (*Mundinger and Taborsky, 2016*). Studying the dynamic regulation and maintenance of pancreatic innervation could provide a better understanding of diabetes etiology and progression.

Whether innervation is a result of vasculature- (*Reinert et al., 2014*; *Cabrera-Vásquez et al., 2009*) or neural crest- (*Plank et al., 2011*; *Nekrep et al., 2008*; *Muñoz-Bravo et al., 2013*; *Kozlova and Jansson, 2009*; *Jiang et al., 2003*; *Young and Newgreen, 2001*; *Arntfield and van der Kooy, 2013*) derived guidance cues remains controversial. And although lineage tracing and knockout studies have suggested a neural crest origin for pancreatic nerves (*Plank et al., 2011*), interpretations can be limited by the Cre driver line specificity (*Chen et al., 2017*; *Lewis et al., 2013*). Additionally, it remains difficult to differentiate between the local effects of autonomic fibers on islet cell physiology from the simultaneous influence on other organs that can indirectly modulate hormone secretion (*Taborsky, 2011*). The zebrafish, with organs homologous to mammalian ones and conserved signaling and metabolic pathways (*Schlegel and Stainier, 2007*; *Seth et al., 2013*; *Ober et al., 2003*), is ideal to study the role of neuronal innervation during endocrine pancreas development. The transparency and rapid embryogenesis of zebrafish, together with fluorescent reporters, allow for the study of developmental processes with single-cell resolution using in vivo time-lapse imaging (*Beis and Stainier, 2006*). Insights into the innervation process could provide clues on the mechanisms utilized by autonomic innervation to regulate islet cell function, growth and survival.

## Results

### Pancreatic innervation in zebrafish is established early in development

Unlike most other vertebrate models, the zebrafish develops externally, and its transparency during embryonic and early larval stages makes it highly attractive to study tissue organogenesis using in vivo time-lapse imaging. To understand pancreatic innervation in zebrafish, we first determined an approximate timeframe of innervation by conducting whole mount immunostaining starting at 50 hr post fertilization (hpf). We found that the onset of pancreatic innervation arises early in development (*Figure 1A–D*), preceding the fusion of the dorsal and ventral pancreatic buds (*Field et al., 2003*), and that prior to 120 hpf, the majority of this innervation is derived from the vagus nerve (*Figure 1A*). Looking more closely, we first observed the migration of the vagus nerve near the pancreatic islet at 50 hpf (*Figure 1B*), parasympathetic extensions from the vagus nerve towards the pancreatic islet by 75 hpf (*Figure 1C*), and an extensive network of nerve fibers within the islet by 100 hpf (*Figure 1D*). Both juvenile (25 days post fertilization; dpf) and adult (1 year old) zebrafish display dense innervation networks within the endocrine pancreas (*Figure 1E–F*). By 25 dpf, the celiac ganglion provides sympathetic innervation (*Figure 1E*), as previously described (*Podlasz et al., 2016*), and at this stage intra-pancreatic innervation between the primary and secondary islets is also present (*Figure 1E*).

### In vivo time-lapse imaging reveals new cellular events preceding pancreatic innervation

The upregulation of *elavl3* expression is a hallmark of post-mitotic neuron differentiation (*Dyer et al., 2014*), and the *elavl3* promoter activity can be used as a pan-neuronal reporter (*Stevenson et al., 2012*). The triple transgenic line *Tg(elavl3:GAL4-VP16); Tg(UAS:EGFP-CAAX); Tg(sst2:RFP)* enables in vivo visualization of the innervation process in conjunction with endocrine pancreas development, where delta cells are expressing the RFP reporter. Single cell resolution imaging allowed us to define the key cellular events leading to the establishment of parasympathetic innervation (*Figure 2A–F*). Post-mitotic neurons, with *elavl3* promoter activity, were closely associated with the endocrine islet very early in development (*Figure 2A*). Starting around this time, neurons began to migrate toward the periphery of the endocrine cluster, and from 28 hpf a subset of neurons detached from its anterior portion (*Figure 2A*, *Figure 2—videos 1–2*). Whole mount immunostaining showed that the cells detaching from the endocrine cluster were not alpha, beta, or delta cells (*Figure 2B*). This neuronal detachment was a sequential process, whereby the first observed detachment event occurred at 28–31 hpf and additional neurons detached from 31 to 50 hpf (*Figure 2A, C*). Subsequently, these neurons migrated rostrally, away from the endocrine cluster (*Figure 2C*,

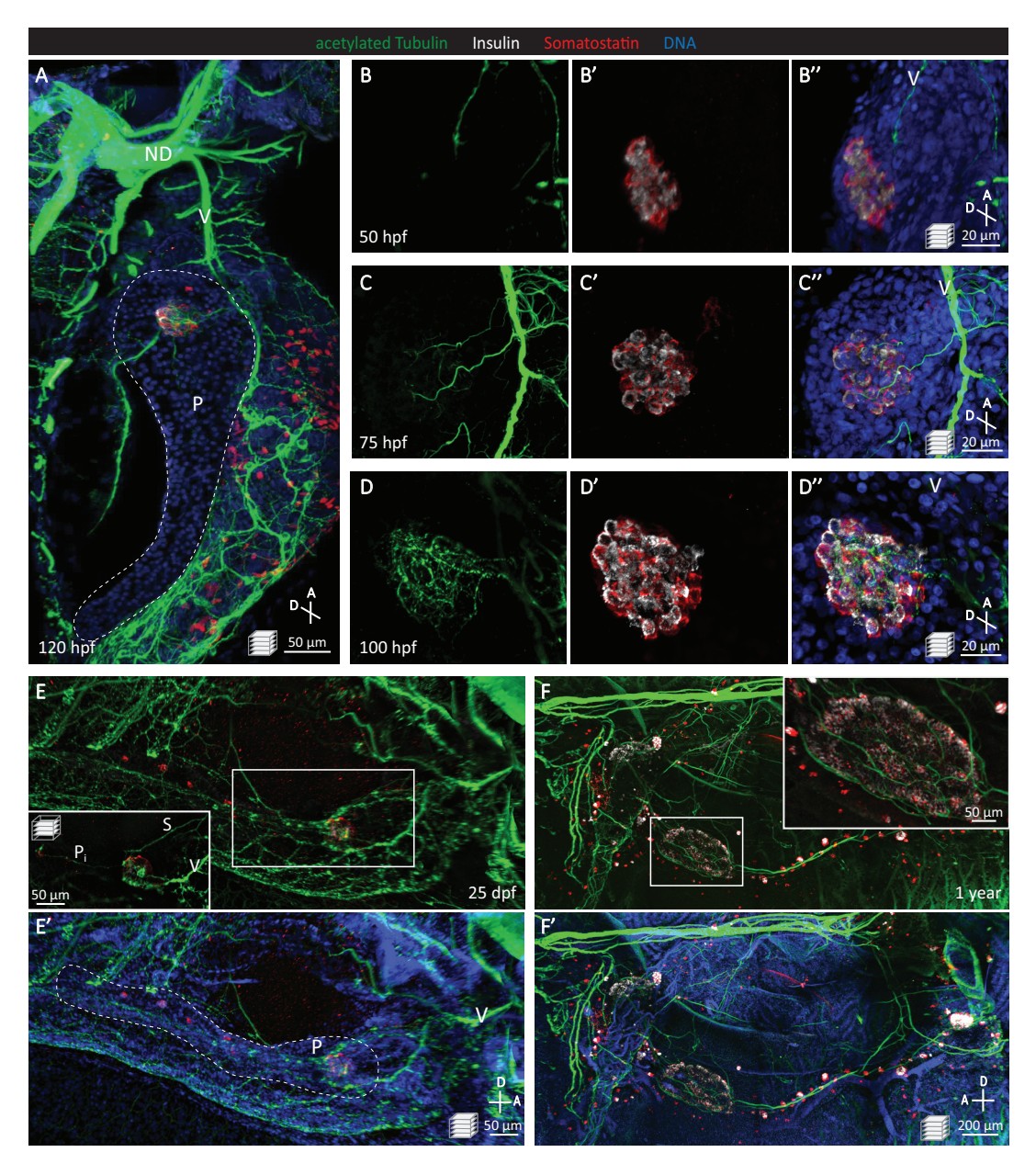

**Figure 1.** Pancreatic islet innervation in zebrafish is established early in development and maintained in juvenile and adult stages. (A-D) Whole mount immunostaining of wild-type zebrafish at 50, 75, 100, 120 hr post fertilization (hpf) for acetylated Tubulin (nerves), Insulin (beta cells), Somatostatin (delta cells), and DAPI (DNA). (E–F) Whole mount immunostaining of 25 days post fertilization (dpf) zebrafish and 1 year zebrafish pancreas and intestine following tissue clearing with the CLARITY protocol. Maximum intensity projections are presented; A, anterior; D, dorsal; V, vagus nerve; ND, nodose ganglion; P, pancreas; S, sympathetic innervation; Pi, intra-pancreatic innervation.

DOI: https://doi.org/10.7554/eLife.34519.002

*Figure 2—video 3*). Next, the vagus nerve extending from the nodose ganglion migrated caudally past the endocrine islet, and the vagus reporter signal overlapped with enteric nerves at 59–65 hpf (*Figure 2D*, *Figure 2—video 4*). As the detached neurons continued to migrate away from the endocrine cells, we could observe neurites extending toward the endocrine cluster (*Figure 2E*, *Figure 2—video 5*). This process was followed by interaction of the detached neurons with the vagus nerve (*Figure 2—video 6*). At 80 hpf, immunostaining revealed the extension of neurites derived

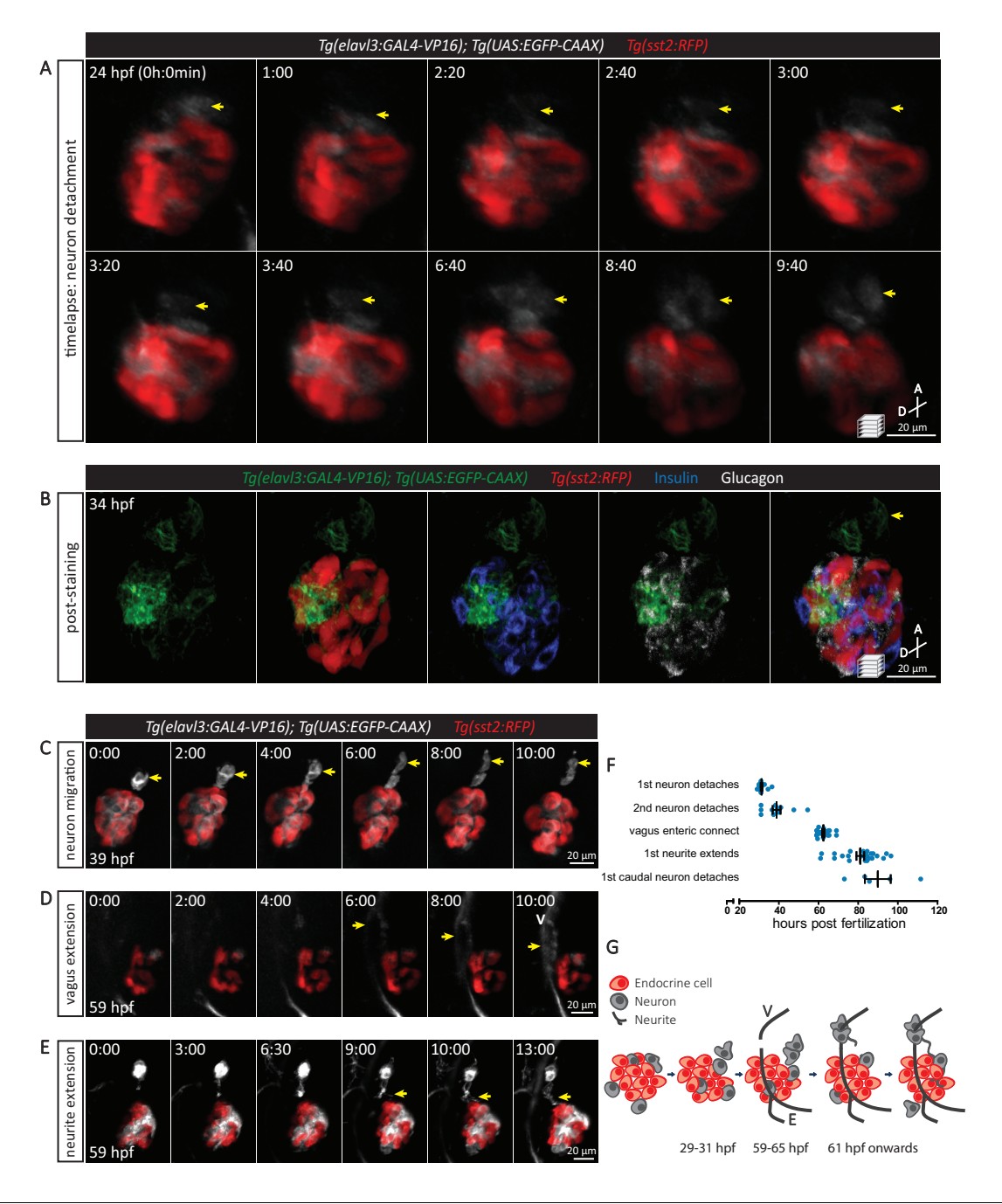

**Figure 2.** The sequence of cellular events preceding pancreatic islet parasympathetic innervation is revealed with single-cell resolution time-lapse confocal imaging. (**A**) *Tg(elavl3:Gal4-VP16); Tg(UAS:EGFP-CAAX); Tg(sst2:RFP)* zebrafish mounted in 0.5% agarose containing 0.017% tricaine were imaged with laser scanning confocal microscopy at 20 min time intervals. Maximum intensity projections of selected timeframes are presented (A, anterior; D, dorsal). Yellow arrows point to the detachment of neurons from the pancreatic islet. (**B**) Whole mount immunostaining at 34 hpf for GFP (neurons), RFP (delta cells), Insulin (beta cells), and Glucagon (alpha cells) after 10 hr time-lapse imaging. The detached neurons are not positive for endocrine cell markers. (**C-E**) Confocal imaging of *Tg(elavl3:Gal4-VP16); Tg(UAS:EGFP-CAAX); Tg(sst2:RFP)* zebrafish at 20–30 min time intervals. Maximum intensity projections of selected timeframes are presented. Yellow arrows point to the cellular events of interest. (**F**) Quantification of the time when the indicated cellular events were observed for individual fish (mean ± SEM). (**G**) Schematic of the sequence of cellular events preceding parasympathetic innervation of the pancreatic islet. V, vagus nerve; E, enteric nerve.

DOI: https://doi.org/10.7554/eLife.34519.003

The following video and figure supplement are available for figure 2:

*Figure 2 continued on next page*

*Figure 2 continued*

**Figure supplement 1.** A subset of pancreatic nerve extensions derives from neurons that were once in close contact with endocrine cells.

DOI: https://doi.org/10.7554/eLife.34519.004

**Figure 2—video 1.** Time-lapse imaging shows detachment of neurons from the pancreatic islet.

DOI: https://doi.org/10.7554/eLife.34519.005

**Figure 2—video 2.** Time-lapse imaging shows detachment of neurons from the pancreatic islet.

DOI: https://doi.org/10.7554/eLife.34519.006

**Figure 2—video 3.** Time-lapse imaging shows neurons migrating away from the pancreatic islet.

DOI: https://doi.org/10.7554/eLife.34519.007

**Figure 2—video 4.** Time-lapse imaging shows vagus nerve extending past the pancreatic islet.

DOI: https://doi.org/10.7554/eLife.34519.008

**Figure 2—video 5.** Time-lapse imaging shows neurite extending toward the pancreatic islet.

DOI: https://doi.org/10.7554/eLife.34519.009

**Figure 2—video 6.** Time-lapse imaging shows interaction of detached neurons with the vagus nerve.

DOI: https://doi.org/10.7554/eLife.34519.010

**Figure 2—video 7.** Time-lapse imaging shows detachment and caudal migration of neurons from the pancreatic islet.

DOI: https://doi.org/10.7554/eLife.34519.011

from *elavl3*-positive cells toward the islet (*Figure 2—figure supplement 1*). Interestingly, neurons also detached from the posterior end of the endocrine cluster and migrated caudally toward the developing secondary islets (*Figure 2—video 7*), while the other neurons stayed on the periphery of the primary islet. These latter two neuronal populations could respectively be important for the formation of intra-pancreatic innervation (*Figure 1E*) and contribute to the peri-endocrine nerve plexus that has been reported in rodents (*Ushiki and Watanabe, 1997*). Due to the high degree of variation in vagus innervation, which was also observed for enteric innervation (*Olsson et al., 2008*), and the slight developmental delay that occurs during the time-lapse imaging of anesthetized zebrafish, we observed variability in the exact developmental timing of individual cellular events in different embryos (*Figure 2F*). Nonetheless, the order of the cellular events was always the same and in combination with the immunostaining data, it allows us to propose a model for the initial establishment of parasympathetic innervation (*Figure 2G*).

A neural crest origin for pancreatic nerves has been suggested from lineage tracing and knockout studies in mouse (*Plank et al., 2011*). Using lineage tracing with *Tg(sox10:CreERT2, myl7:GFP); Tg(ubb:loxP-CFP-loxP-nuc-mCherry)* zebrafish upon tamoxifen treatment from 16 to 24 hpf (when the neural crest population reaches its peak) (*Mongera et al., 2013*), we found a population of neural-crest-derived cells expressing the *elavl3* promoter in association with the endocrine islet at 35 hpf (*Figure 3A,B*). Live imaging of *Tg(sox10:GAL4); Tg(UAS:GFP); Tg(sst2:RFP)* zebrafish revealed the migration of neural-crest-derived cells toward the islet before 27 hpf, their contact with the islet, and subsequently their migrating away from the islet (*Figure 3C*). Immunostaining showed that the cells migrating away from the islet were not beta or delta cells (*Figure 3D*). Although *elavl3* promoter activity was observed in a subpopulation of endocrine cells (*Figure 2B*), our data suggest that they do not contribute to the subset of cells migrating away from the islet. Interestingly, we also did not observe immunostaining for HuC/HuD in this subpopulation of neural-crest-derived cells (*Figure 3E*), suggesting that these cells are not yet fully mature neurons.

To further test whether the pancreatic nerve density was indeed neural crest derived, we lineage-traced neural crest cells with *Tg(sox10:CreERT2, myl7:GFP); Tg(ubb:loxP-CFP-loxP-nuc-mCherry)* zebrafish upon tamoxifen treatment from 16 to 24 hpf (*Figure 4A–E*). The neural-crest-derived cells were not positive for Insulin, Glucagon, or Somatostatin staining at 90 hpf (*Figure 4B*). We found that nerve extensions toward the primary islet at 90 and 120 hpf were indeed derived from cells that were once positive for *sox10* promoter activity (*Figure 4C–E*). We also observed *sox10*-derived cells that remained on the periphery of the primary islet (*Figure 4C–E*). The decrease in the number of *sox10*-derived cells in between the vagus nerve and pancreatic islet when comparing 90 to 120 hpf suggests that these cells are migrating toward the vagus nerve (*Figure 4C–E*). Additionally, immunostaining of *sox10* mutants, which exhibit a severe depletion in neural crest cells (*Dutton et al., 2001*), revealed a complete absence of pancreatic innervation even though the vagus nerve extension was present (*Figure 4—figure supplement 1*). Overall, these data agree with the neural crest

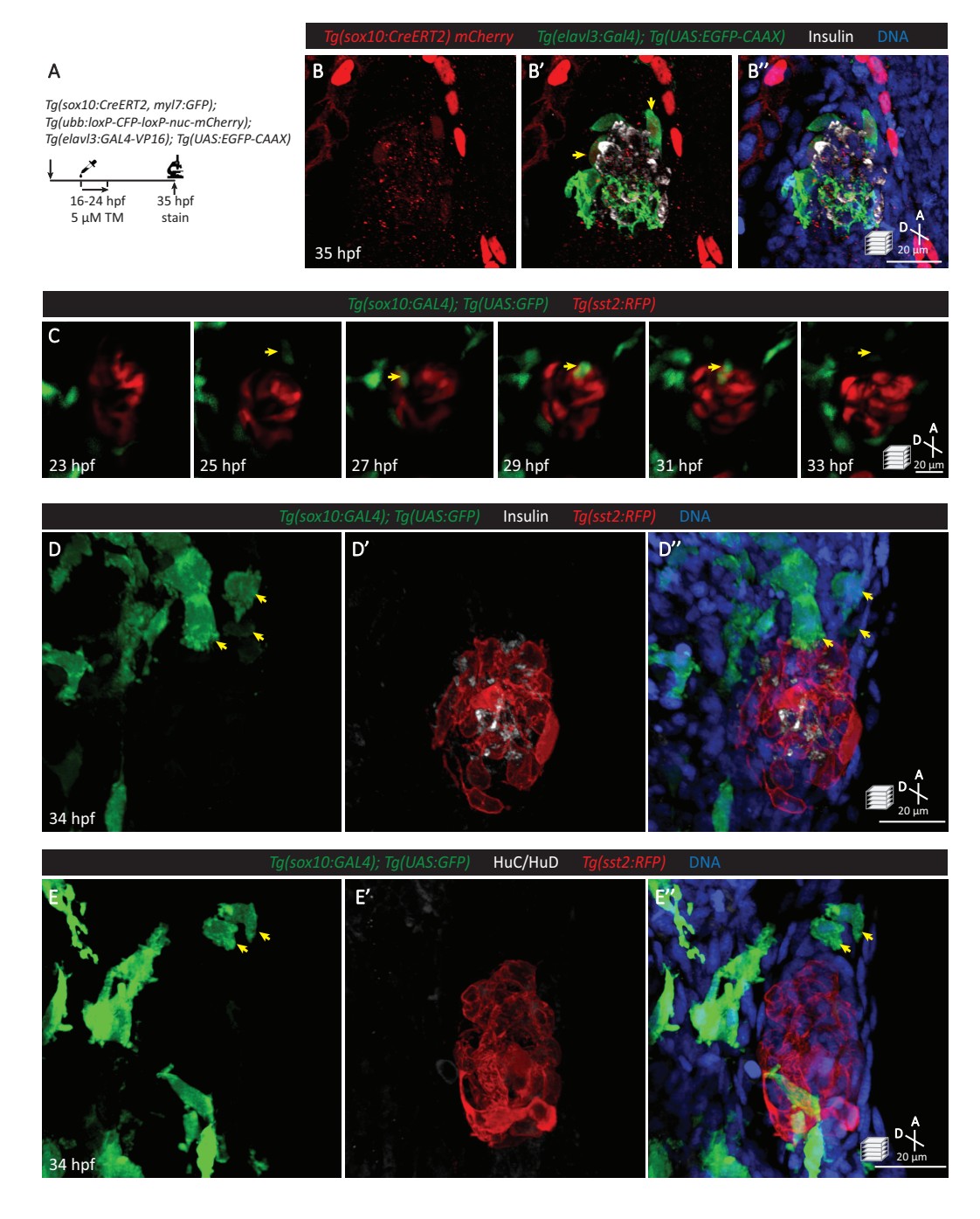

**Figure 3.** Neural crest cells are in close contact with pancreatic endocrine cells early in development. (A) Lineage tracing of neural crest cells in *Tg (sox10:CreERT2, myl7:GFP); Tg(ubb:loxP-CFP-loxP-nuc-mCherry); Tg(elavl3:GAL4-VP16); Tg(UAS:EGFP-CAAX)* zebrafish following 5 µM tamoxifen treatment from 16 to 24 hpf and staining at 35 hpf. (B) Whole mount immunostaining at 35 hpf for mCherry (neural-crest-derived cells), GFP (*elavl3*-positive cells), Insulin (beta cells), and DAPI (DNA). (C) Confocal imaging of *Tg(sox10:GAL4); Tg(UAS:GFP); Tg(sst2:RFP)* zebrafish mounted in 0.5% agarose from 23 to 33 hpf. Yellow arrow points to a neural crest cell in close proximity to endocrine pancreatic cells and briefly contacting islet cells before migrating away. (D) Whole mount immunostaining at 34 hpf for GFP (*sox10*-positive cells), Insulin (beta cells), RFP (delta cells), and DAPI (DNA). Yellow arrows point to neural-crest-derived cells that were once in contact with the pancreatic islet. (E) Whole mount immunostaining at 34 hpf for GFP (*sox10*-positive cells), HuC/HuD (mature neurons), RFP (delta cells), and DNA (DNA). Yellow arrows point to neural-crest-derived cells that were once in contact with the pancreatic islet.

DOI: https://doi.org/10.7554/eLife.34519.012

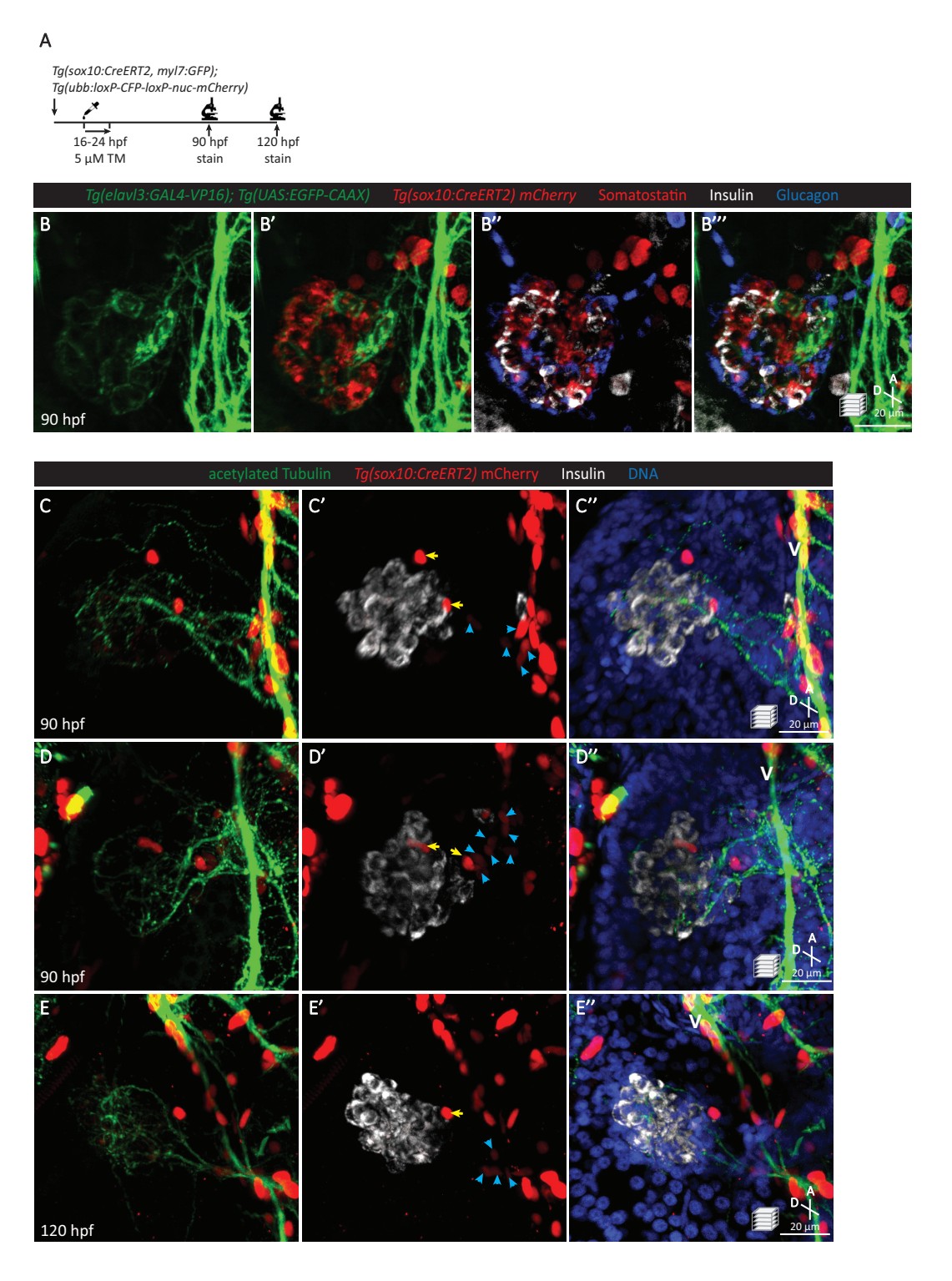

**Figure 4.** Pancreatic islet parasympathetic innervation is derived from the neural crest. (**A**) Lineage tracing of neural crest cells in *Tg(sox10:CreERT2, myl7:GFP)*; *Tg(ubb:loxP-CFP-loxP-nuc-mCherry)* zebrafish following 5 μM tamoxifen treatment from 16 to 24 hpf and staining at 90 and 120 hpf. (**B**) Whole mount immunostaining at 90 hpf for GFP (*elavl3*-positive cells), mCherry (neural-crest-derived cells), Somatostatin (delta cells), Insulin (beta cells), and Glucagon (alpha cells). (**C–E**) Whole mount immunostaining at 90 (**C, D**) and 120 (**E**) hpf for acetylated Tubulin (nerves), Insulin (beta cells), mCherry (neural-crest-derived cells), and DAPI (DNA). Yellow arrows point to neural-crest-derived cells on the periphery of the pancreatic islet; blue arrowheads point to neural-crest-derived cells projecting neural extensions toward the pancreatic islet, and some of these cells are adjacent to the vagus nerve (**V**).

*Figure 4 continued on next page*

*Figure 4 continued*

DOI: https://doi.org/10.7554/eLife.34519.013

The following figure supplement is available for figure 4:

**Figure supplement 1.** Neural crest cells are essential for the establishment of islet parasympathetic innervation.

DOI: https://doi.org/10.7554/eLife.34519.014

origin of pancreatic innervation that has been reported in rodents (*Plank et al., 2011*; *Nekrep et al., 2008*; *Muñoz-Bravo et al., 2013*; *Kozlova and Jansson, 2009*; *Jiang et al., 2003*; *Young and Newgreen, 2001*).

## The vasculature is not essential for the establishment of innervation

Several mechanisms of pancreatic innervation have been proposed, including axons following along blood vessels to eventually innervate the endocrine pancreas (*Reinert et al., 2014*; *Cabrera-Vásquez et al., 2009*), and neural crest cells found closely associated with endocrine cells eventually differentiating into neural and glial cells (*Plank et al., 2011*; *Nekrep et al., 2008*). We carefully examined the cellular events preceding the onset of endocrine pancreas innervation using the zebrafish model. By utilizing *cloche* mutants, which lack most endothelial cells (*Stainier et al., 1995*; *Liao et al., 1997*; *Reischauer et al., 2016*) including those endothelial cells that give rise to the pancreatic vasculature (*Field et al., 2003*), we wanted to examine whether blood vessels were crucial for islet innervation (*Figure 5A*). Even though *cloche* mutants display fairly severe defects (*Figure 5B–D*) and the vasculature was completely absent throughout the gastrointestinal system, as visualized by *Tg(kdrl:*GFP) expression, by 80 hpf both the right and left vagus nerve extensions could be observed in wild-type and mutant fish (*Figure 5E–F*). Interestingly, extensive innervation of the pancreatic islet, comparable to that in wild-type, was also observed in *cloche* mutants (*Figure 5G–I*). A dense network of nerves could still be observed in the pancreatic islet at 4 dpf (*Figure 5—figure supplement 1*), suggesting that this initial stage of islet innervation and its expansion does not require signaling from endothelial cells. Thus, although it is likely that maintenance of nerve density requires vascular supply (*Reinert et al., 2014*), our studies indicate that the presence of a vascular network is not crucial for the establishment of innervation.

## Targeted ablation of specific peri-endocrine neurons results in reduced pancreatic innervation

Our in vivo time-lapse imaging revealed the detachment of neurons from the endocrine islet and their migration toward the vagus nerve. To determine whether this subpopulation of detached neurons plays a crucial role in the innervation density observed at 75 hpf (*Figure 1C*), we conducted targeted cell ablation studies (*Figure 6A*). We utilized a two-photon laser to specifically ablate at 31–33 hpf neurons that have detached from the developing islet (*Figure 6C* mock ablation as control; *Figure 6D* two-photon ablation) and conducted immunostaining at 80 hpf, when extensive parasympathetic innervation is already observed in controls (*Figure 6E*). In comparison to the mock ablated controls, where cells immediately adjacent to the leading front of the detached neurons were targeted with the same laser intensity (*Figure 6C*), there was a significant reduction in innervation density upon ablation of the detached cluster of neurons (*Figure 6F–G*), while extension of the vagus nerve and enteric innervation appeared unaffected (*Figure 6F*). The absence of differences in size and gross morphology of the fish suggests that this decline in innervation density was not due to developmental delay (*Figure 6B*). Given that targeted ablation did not disrupt development of the enteric nervous system, we next wanted to determine whether endocrine cell development was perturbed upon disruption of islet nerve density. Immunostaining at 80 hpf for Insulin and Somatostatin revealed no significant differences in islet mass or beta cell numbers between the ablated fish and mock ablated controls; however, a significant decrease in delta cell numbers was observed (*Figure 6H–J*), suggesting that early innervation is involved in establishing or maintaining the delta cell pool.

In order to determine whether synaptic transmission was required for parasympathetic innervation, we overexpressed botulinum toxin light chain B in post-mitotic neurons using *Tg(elavl3:GAL4-VP16); Tg(UAS:BoTxBLC-GFP)* fish (*Figure 7A*). Inhibition of neural activity did not result in

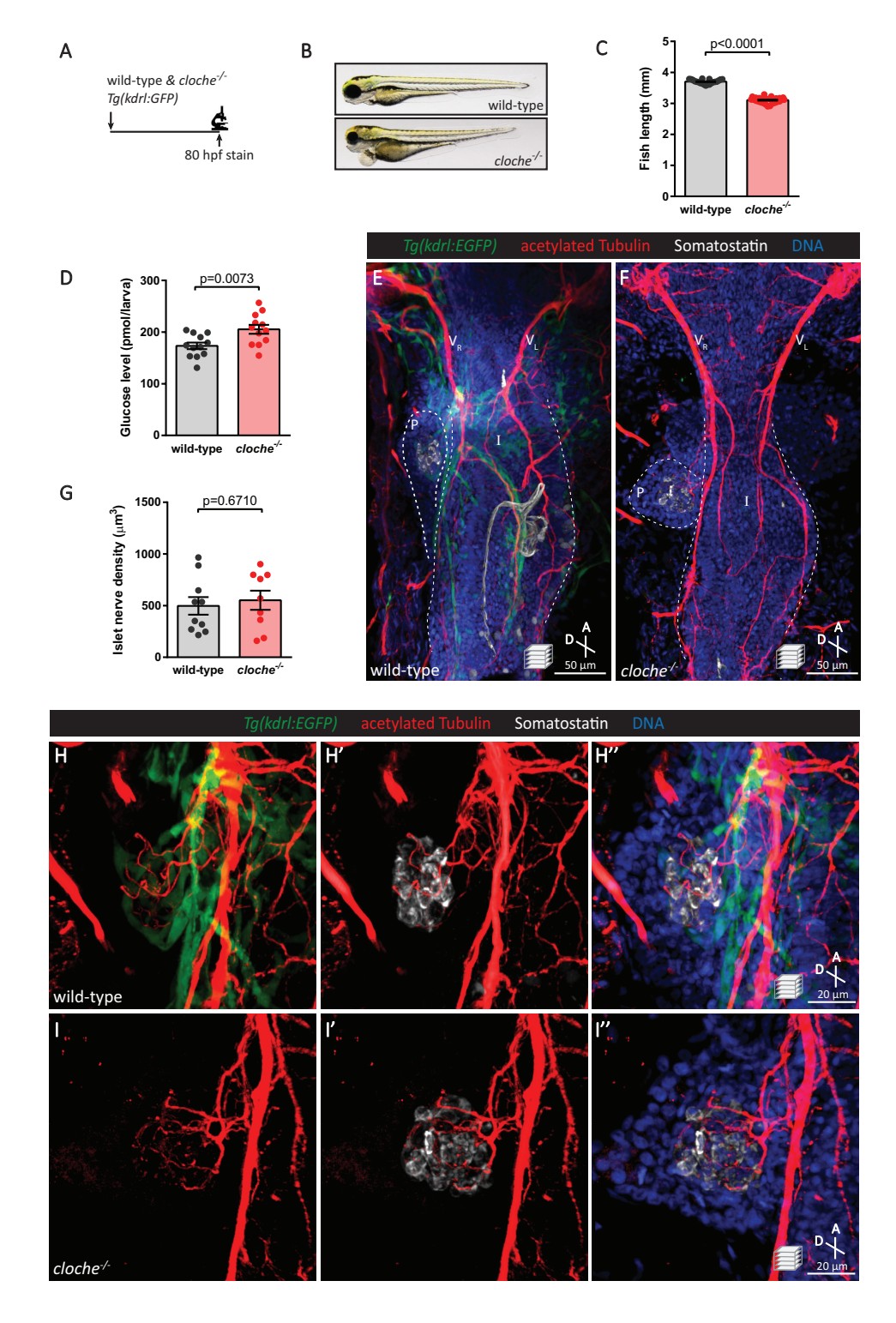

**Figure 5.** The vasculature is not essential for the initial establishment of pancreatic islet parasympathetic innervation. (A) Immunostaining analysis of innervation density in *Tg(kdrl:GFP)* wild-type and *cloche* mutants at 80 hpf. (B) Wild-type and *cloche*[-/-] larvae were imaged at 80 hpf. (C) Body length measurements at 80 hpf, mean ± SEM, n = 21–27 animals. (D) Whole larva free glucose level measurements at 80 hpf, mean ± SEM, n = 12 batches of five larvae per replicate. (E–I) Whole mount immunostaining at 80 hpf for GFP (blood vessels), acetylated Tubulin (nerves), Somatostatin (delta cells), and DAPI (DNA). Maximum intensity projections are presented; A, anterior; D, dorsal. Quantification of islet nerve density (G), mean ± SEM, n = 9–10

*Figure 5 continued on next page*

*Figure 5 continued*

animals, p-values from t tests are presented. No significant difference in vagus nerve extension and islet innervation was observed between wild-type and *cloche*$^{-/-}$ larvae. V$_R$, right vagus nerve; V$_L$, left vagus nerve; P, pancreas; I, intestine.

DOI: https://doi.org/10.7554/eLife.34519.015

The following figure supplement is available for figure 5:

**Figure supplement 1.** The vasculature is not essential for the initial establishment of islet parasympathetic innervation.

DOI: https://doi.org/10.7554/eLife.34519.016

developmental delay at 80 hpf (*Figure 7B–C*); however, a significant decrease in islet nerve density was observed even though the vagus nerve developed normally (*Figure 7E–I*). Similar to the targeted cell ablation approach, analysis of immunostained fish revealed no differences in beta cell numbers but a significant decrease in delta cell numbers (*Figure 7K–L*). Interestingly, delta cell hypertrophy was observed (*Figure 7—figure supplement 1*), which could account for the significant increase in primary islet mass (*Figure 7J*). Elevated free glucose levels were also observed upon inhibition of neural transmission at 80 hpf (*Figure 7D*), and they remained elevated at 5 dpf, similar to what we observed in *sox10* mutants which lack neural crest cells (*Figure 7—figure supplement 2*). Whether the increase we observed in whole larva free glucose levels is directly related to pancreatic islet development or perturbed functional modulation by parasympathetic innervation on the islet or other peripheral organs remains to be examined. To further analyze the changes in endocrine cell mass, we used a genetic ablation method (*Curado et al., 2007*; *Pisharath and Parsons, 2009*) whereby Nitroreductase (NTR) was specifically expressed in post-mitotic neurons in *Tg(elavl3:GAL4-VP16)*; *Tg(UAS:NTR-mCherry)* fish (*Figure 7—figure supplement 3*). Upon treatment with metronidazole (MTZ, a pro-drug that is converted to a cytotoxic form in the presence of NTR) from 24 to 80 hpf to induce global neuronal ablation, we detected significant decreases in beta and delta cell numbers (*Figure 7—figure supplement 3C–D*). Overall, our data suggest that the early establishment of parasympathetic innervation requires the presence of neurons that are initially associated with the endocrine pancreas as well as active neural transmission. Notably, significant changes in pancreatic delta cell mass were observed upon decreased islet nerve density.

## Discussion

Autonomic innervation can regulate the function of the endocrine pancreas (*Ahrén, 2000*). The brain centers linked to pancreatic innervation have been mapped with retrograde neuronal tracing (*Rosario et al., 2016*; *Kreier et al., 2006*). More recently, studies have demonstrated that modulation of nerve activity in the brain can influence pancreas function (*Kume et al., 2016*; *Croizier et al., 2016*). However, our knowledge of the local effects of nerves is limited due to the effects of nerve transmission on other target organs. Through our studies of neuronal development in zebrafish, we identified a subpopulation of neurons that are involved in pancreas-specific parasympathetic innervation, and ablation of these neurons significantly diminished local nerve density. Further lineage-tracing studies would be useful to distinguish nerves coming from the detached neurons to those entirely from the vagus. Nonetheless, this targeted ablation method provided us with a model whereby nerve density remains unaltered in other tissues and allowed us to interrogate the local effects of nerves on endocrine pancreas development. Interestingly, a specific effect on delta cell mass was observed upon diminished nerve density. Likewise, global neural inhibition and global neuron ablation also led to a decline in delta cell mass. However, in the case of global neuron ablation, we also observed a decline in beta cell mass likely due to secondary effects from the extensive levels of neuronal cell death. Given the potential role of Somatostatin signaling in mediating migration of developing neurons (*Yacubova and Komuro, 2002*), it is possible that delta cell loss could worsen the decline in innervation density.

Similar to what has been reported in rodents (*Hsueh et al., 2017*; *Tang et al., 2018*), a dense supply of nerve fibers could be observed within the endocrine pancreas of developing and adult zebrafish. Parasympathetic innervation predominates prior to 5 dpf, but by 25 dpf, sympathetic innervation extending from the celiac ganglia was also detected. Galaninergic nerves have been previously identified in the zebrafish pancreas (*Podlasz et al., 2016*), and as in non-human primates these nerves are parasympathetic (*Verchere et al., 1996*). The neurons present on the periphery of

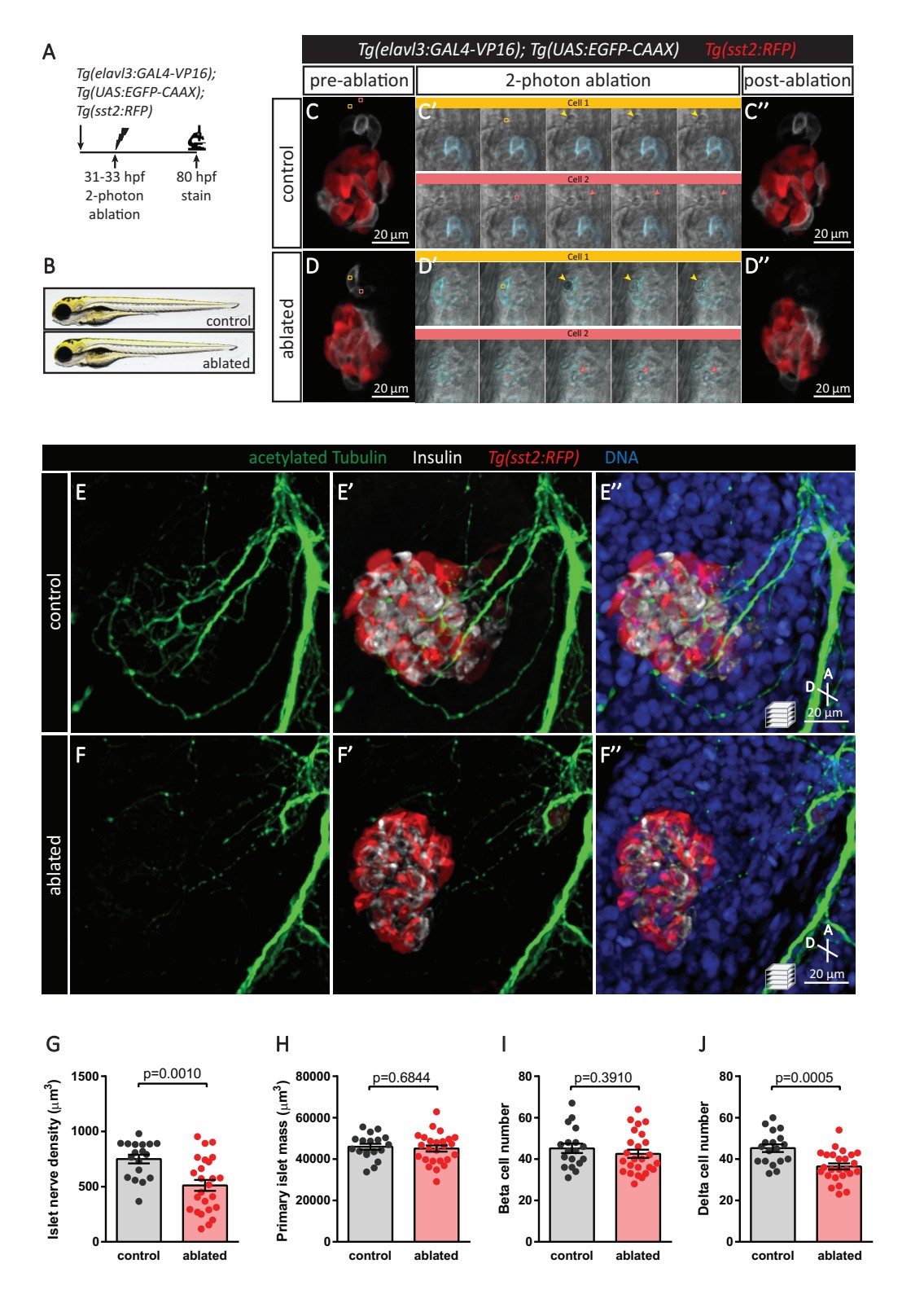

**Figure 6.** Targeted ablation studies reveal the crucial role of peri-islet neurons for the initiation of pancreatic islet parasympathetic innervation. (**A**) Schematic of the two-photon ablation experiment. (**B**) Gross morphology at 80 hpf was comparable between control and ablated fish. (**C–D**) *Tg(elavl3: Gal4-VP16); Tg(UAS:EGFP-CAAX); Tg(sst2:RFP)* zebrafish mounted in 0.5% agarose with tricaine were subjected to two-photon laser ablation. The
*Figure 6 continued on next page*

*Figure 6 continued*
detached neuron clusters were ablated between 31 and 33 hpf. The control was mock ablation of cells adjacent to the leading edge of the migrating neurons using the same laser intensity. Pre-ablation, short time-lapse immediately following ablation, and post-ablation images are displayed. Orange and pink boxes outline the regions of ablation; arrowheads point to ablated cells. (E–J) Whole mount immunostaining at 80 hpf for acetylated Tubulin (nerves), Insulin (beta cells), RFP (delta cells), and DAPI (DNA). Quantification of islet nerve density (G), primary islet mass (H), beta cell number (I), and delta cell number (J), mean ± SEM, n = 18–25 animals, p-values from t tests are presented.
DOI: https://doi.org/10.7554/eLife.34519.017

the islet as revealed in our studies could give rise to the peri-endocrine nerve plexus which has been observed in rodents (*Tang et al., 2014*; *Ushiki and Watanabe, 1997*). Targeted ablation of these neurons at later stages should help decipher their role.

To our knowledge, ours is the first report of the physical proximity of neurons and their targets prior to the migration of these neurons away from the developing target organ and eventual organ innervation. The power of in vivo time-lapse imaging and targeted cell ablation allowed us to characterize the cellular events leading to parasympathetic innervation of the endocrine pancreas in zebrafish. The uncovered sequence of events is reminiscent of the pioneer neuron model first proposed over a century ago and which has been observed in various model systems, including *Drosophila* (*Sánchez-Soriano and Prokop, 2005*; *Lin et al., 1995*; *Jacobs and Goodman, 1989*), zebrafish (*Hoijman et al., 2017*; *Wanner and Prince, 2013*), and mouse (*Morante-Oria et al., 2003*; *Stainier and Gilbert, 1990*). However, a key difference between what we observed and the classical model is that in the zebrafish pancreas these neurons are not migrating toward the target organ where they provide instructional cues for neuronal specification (*Hoijman et al., 2017*) or lay down axon tracks for the following neurons (*Wanner and Prince, 2013*). Instead, we observed the migration of these neurons away from the target organ while maintaining contact via neural extensions, which may subsequently provide tracks for other neurons to migrate along to reach their targets. Indeed, the diminished innervation density following their targeted ablation revealed the crucial role of these neurons that initially associate with the endocrine cells. Future studies designed to identify the guidance signals regulating the directional migration of these subpopulation of neurons and investigate the effect of the lack of pancreas innervation on islet cell development and regeneration should lead to further understanding of the innervation process and its importance in development and disease. In addition, it will be interesting to examine in detail the innervation of other organs to determine whether this model of early local neuronal differentiation followed by migration away from the target is a more general phenomenon.

## Materials and methods

### Zebrafish transgenic lines and husbandry

All zebrafish husbandry was performed under standard conditions in accordance with institutional (MPG) and national ethical and animal welfare guidelines. Adult zebrafish were fed a combination of fry food (Special Diet Services) and brine shrimp five times daily and maintained under a light cycle of 14 hr light: 10 hr dark at 28.5°C. Transgenic and mutant lines used were as described (*Table 1*): Tg(elavl3:GAL4-VP16)$^{zf357}$ (*Stevenson et al., 2012*) (promoter activity in post-mitotic neurons), Tg (UAS:EGFP-CAAX)$^{m1230}$ (*Fernandes et al., 2012*), Tg(sst2:RFP)$^{gz19}$ (*Li et al., 2009*) (promoter activity in pancreatic delta cells), Tg(UAS:BoTxBLC-GFP)$^{icm21}$ (*Sternberg et al., 2016*), cloche$^{s5}$ (*Liao et al., 1997*), cloche$^{m39}$ (*Stainier et al., 1995*), Tg(kdrl:EGFP)$^{s843}$ (*Jin et al., 2005*) (promoter activity in endothelial cells), Tg(sox10:GAL4)$^{sq9}$ (*Lee et al., 2013*) (promoter activity in neural crest cells), Tg (sox10:CreERT2, myl7:GFP)$^{t007}$ (*Mongera et al., 2013*), Tg(ubb:loxP-CFP-loxP-nuc-mCherry)$^{jh63}$ (*Wang et al., 2015*) (ubiquitous promoter activity), sox10$^{tw2}$ (*Kelsh et al., 1996*), Tg(UAS:NTR-mCherry)$^{c264}$ (*Davison et al., 2007*). Whole larva glucose levels were measured with Glucose Assay Kit (Millipore) from five larvae per replicate.

### In vivo confocal microscopy

Live zebrafish between 1 and 5 dpf were anesthetized with 0.017% Tricaine and mounted in 0.5% low melting agarose in egg water containing 0.017% Tricaine for confocal imaging. Zeiss LSM780

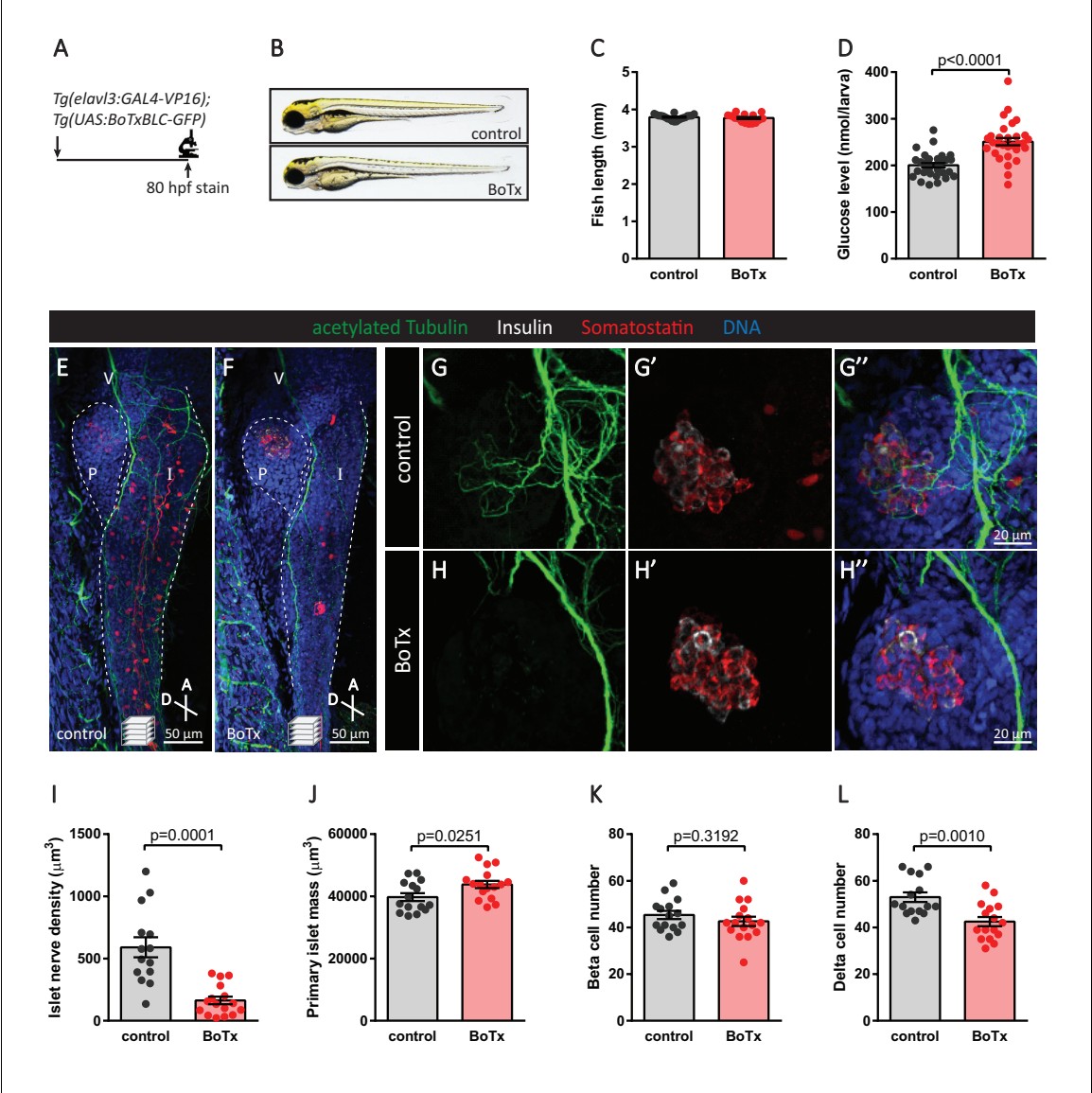

**Figure 7.** Inhibition of neural synaptic transmission diminishes islet parasympathetic innervation. (**A**) Botulinum toxin (BoTx) was expressed in post-mitotic neurons to inhibit neurotransmitter release in *Tg(elavl3:Gal4-VP16); Tg(UAS:BoTxBLC-GFP)* zebrafish. (**B**) Control and BoTx-positive fish were imaged at 80 hpf. (**C**) Body length measurements at 80 hpf, mean ± SEM, n = 17–20 animals. (**D**) Whole larva free glucose level measurements at 80 hpf, mean ± SEM, n = 30 batches of five larvae per replicate. (**E–L**) Whole mount immunostaining at 80 hpf for acetylated Tubulin (nerves), Insulin (beta cells), Somatostatin (delta cells), and DAPI (DNA). Maximum intensity projections are presented; A, anterior; D, dorsal; V, vagus nerve; P, pancreas; I, intestine. Quantification of islet nerve density (**I**), primary islet mass (**J**), beta cell number (**K**), and delta cell number (**L**), mean ± SEM, n = 14–16 animals, p-values from t tests are presented.

DOI: https://doi.org/10.7554/eLife.34519.018

The following figure supplements are available for figure 7:

**Figure supplement 1.** Inhibition of synaptic transmission results in delta cell hypertrophy.

DOI: https://doi.org/10.7554/eLife.34519.019

**Figure supplement 2.** Diminished nerve density and impaired neural output result in hyperglycemia.

DOI: https://doi.org/10.7554/eLife.34519.020

**Figure supplement 3.** Genetic ablation of neurons decreases pancreatic endocrine cell numbers.

DOI: https://doi.org/10.7554/eLife.34519.021

**Table 1.** List of zebrafish transgenic and mutant lines.

| Name | Specificity/Purpose/ Phenotype | Reference |
|---|---|---|
| *Tissue specific promoter lines* | | |
| *Tg(elavl3:GAL4-VP16)*[zf357] | Post-mitotic neurons | (*Stevenson et al., 2012*) |
| *Tg(sst2:RFP)*[gz19] | Pancreatic delta cells | (*Li et al., 2009*) |
| *Tg(sox10:GAL4)*[sq9] | Neural crest cells | (*Lee et al., 2013*) |
| *Tg(sox10:CreERT2, myl7:GFP)*[t007] | Neural crest cells, heart marker | (*Mongera et al., 2013*) |
| *Tg(ubb:loxP-CFP-loxP-nuc-mCherry)*[jh63] | Ubiquitous | (*Wang et al., 2015*) |
| *Tg(kdrl:EGFP)*[s843] | Endothelial cells | (*Jin et al., 2005*) |
| *UAS lines* | | |
| *Tg(UAS:EGFP-CAAX)*[m1230] | Visualize neurons | (*Fernandes et al., 2012*) |
| *Tg(UAS:BoTxBLC-GFP)*[icm21] | Inhibit neurotransmitter release | (*Sternberg et al., 2016*) |
| *Tg(UAS:NTR-mCherry)*[c264] | Ablate neurons | (*Davison et al., 2007*) |
| *Tg(UAS:GFP)*[zf82] | Visualize neural crest cells | (*Asakawa et al., 2008*) |
| *Mutant lines* | | |
| *cloche*[s5], *cloche*[m39] | Lacking endothelial cells | (*Stainier et al., 1995*; *Liao et al., 1997*) |
| *sox10*[tw2] | Lacking neural crest cells | (*Kelsh et al., 1996*) |

DOI: https://doi.org/10.7554/eLife.34519.022

and LSM880 upright laser scanning confocal microscopes equipped with a Plan-Apochromat 20x/ NA1.0 dipping lens were used for imaging. Time-lapse experiments were conducted in 28.5°C conditions and z-stacks were taken at 20–45 min intervals for overall timeframes of 15–20 hr. Images were analyzed using Imaris software (Bitplane).

## Two-photon laser ablations

Embryos were anesthetized with 0.017% tricaine and mounted in 0.5% agarose. A Chameleon Vision II Ti:Sapphire Laser (Coherent) mounted on a Zeiss LSM880 microscope was used for two-photon single-cell laser ablations. The tunable laser was set at 800 nm to scan an ablation area of 4 $\mu m^2$ at a scan speed of 1 with 10 iterations. Following ablation, embryos were removed from the agarose, raised to 80 hpf, and fixed with 4% paraformaldehyde for subsequent immunostaining.

## Wholemount immunostaining

Zebrafish were euthanized with tricaine overdose prior to overnight fixation in 4% paraformaldehyde dissolved in PBS containing 120 $\mu M$ $CaCl_2$ and 4% sucrose, pH7.4. The skin was manually removed with forceps, without disturbing the internal organs and the zebrafish were permeabilized with 0.5% TritonX-100 for 3 hr at room temperature. Following blocking with 5% donkey serum (Jackson Immunoresearch) in blocking buffer (Dako), samples were incubated in primary antibodies overnight at 4°C, washed 4x with 0.025% TritonX-100 containing PBS, incubated in secondary antibodies overnight at 4°C, washed 4x, incubated in an increasing glycerol gradient of 25, 50, and 75%, and mounted in VectorShield mounting medium. The following antibodies and dilutions were used: guinea pig anti-Insulin polyclonal (1:100, Thermo), rabbit anti-Somatostatin (1:100, BioRad), chicken anti-GFP (1:200, Aves), mouse anti-acetylated Tubulin (1:200, Sigma), mouse monoclonal anti-HuC/D (1:100, Cell Signaling). Secondary antibodies (Jackson ImmunoResearch) used in this study include donkey anti-guinea pig AlexaFluor647 and 405, donkey anti-rabbit Cy3 and AlexaFluor488, donkey anti-mouse AlexaFluor488 and 647, donkey anti-chicken AlexaFluor488. Nuclei were stained with 25 $\mu g/ml$ DAPI. Images were taken on Zeiss LSM700 or LSM800 laser scanning confocal microscopes equipped with a 25x/NA0.8 objective.

## Data analysis

Image data were analyzed using Imaris (Bitplane) and Huygens (Scientific Volume Imaging) softwares. Statistical analysis was performed using Prism (GraphPad).

## Acknowledgements

These studies were supported by funds from the Max Planck Society to DYRS. KK was supported by a NBRP grant from AMED. YHCY was supported by a CIHR Postdoctoral Fellowship, an EMBO Long-Term Fellowship, an HFSP Long-Term Fellowship, and a NIG-JOINT funding. The referenced transgenic lines were generously provided by Zhiyuan Gong, Wolfgang Driever, Michael Parsons, and Christiane Nüsslein-Volhard. We thank Radhan Ramadass for imaging advice.

## Additional information

### Competing interests

Didier YR Stainier: Senior editor, *eLife*. The other authors declare that no competing interests exist.

### Funding

| Funder | Grant reference number | Author |
| --- | --- | --- |
| Max-Planck-Gesellschaft | Open-access funding | Didier YR Stainier |
| Human Frontier Science Program | Long-Term Fellowship | Yu Hsuan Carol Yang |
| European Molecular Biology Organization | Long-Term Fellowship | Yu Hsuan Carol Yang |
| Canadian Institutes of Health Research | CIHR Fellowship | Yu Hsuan Carol Yang |
| Japan Agency for Medical Research and Development | NBRP | Koichi Kawakami |
| National Institute of Genetics | NIG-JOINT Collaborative Research (A2) | Yu Hsuan Carol Yang |

The funders had no role in study design, data collection and interpretation, or the decision to submit the work for publication.

### Author contributions

Yu Hsuan Carol Yang, Conceptualization, Data curation, Formal analysis, Funding acquisition, Validation, Investigation, Visualization, Methodology, Writing—original draft, Project administration, Writing—review and editing; Koichi Kawakami, Resources, Methodology, Writing—review and editing, provided the Tg(UAS:BoTxBLC-GFP) line; Didier YR Stainier, Resources, Supervision, Funding acquisition, Writing—original draft, Writing—review and editing

### Author ORCIDs

Yu Hsuan Carol Yang (iD) http://orcid.org/0000-0001-6663-0302
Koichi Kawakami (iD) https://orcid.org/0000-0001-9993-1435
Didier YR Stainier (iD) http://orcid.org/0000-0002-0382-0026

### Ethics

Animal experimentation: All zebrafish husbandry was performed under standard conditions in accordance with institutional (MPG) and national ethical and animal welfare guidelines approved by the ethics committee for animal experiments at the Regierungspräsidium Darmstadt, Germany (permit numbers B2/1138 and B2/Anz. 1007).

Decision letter and Author response
Decision letter https://doi.org/10.7554/eLife.34519.025
Author response https://doi.org/10.7554/eLife.34519.026

## Additional files

### Supplementary files
• Transparent reporting form
DOI: https://doi.org/10.7554/eLife.34519.023

### Data availability
All data generated/analysed during this study are included in the manuscript. Individual replicates along with the means +/- SEM are plotted for all numerical data in the figures.

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
