## [Decision Letter]

Thank you for submitting your article "A new mode of organ innervation revealed by live imaging of the developing zebrafish pancreas" for consideration by *eLife*. Your article has been reviewed by three peer reviewers, and the evaluation has been overseen by a Reviewing Editor and K VijayRaghavan as the Senior Editor. The following individual involved in review of your submission has agreed to reveal his identity:: Mehboob Hussain (Reviewer #1).

The reviewers have discussed the reviews with one another and the Reviewing Editor has drafted this decision to help you prepare a revised submission.

In general, there was a lot of enthusiasm for this interesting work which will enhance our understanding of pancreatic islet innervation, an area that is currently not well understood. However, despite their enthusiasm, there are a number of issues that the reviewers agree need to be resolved before this work can be considered for publication. Foremost among these is that your study does not reveal the origin of the Elav-expressing cells whose development you have described. Although you show that neural crest is required for pancreatic islet innervation, your study does not provide evidence that these Elav cells are neural crest derived, and thus does not rule out the possibility that they have an endocrine origin. Thus, you will need to investigate both whether neural crest cells and whether endocrine cells from the pancreas contribute to this Elav-expressing population. Sox10 may not be expressed early enough to be definitive for this experiment, thus you may need to use other transgenic lines to look at earlier developmental stages. This analysis will also provide important information on the timing of association of these cells with the developing pancreas.

There are a number of other issues that should also be addressed in your resubmission.

1) We suggest changing the title to reflect the fact that you only investigate pancreatic islet innervation.

2) It would be useful to distinguish neurites derived from the vagus nerve from those of the Elav-expressing cells.

3) It is difficult to see the detachment of the Elav-expressing cells in your movies, making it unclear whether the later cells arise from proliferation or cells that turn on Elav expression at a later time.

4) The laser-directed, ntr-directed, and botulinic toxin-directed ablations have different outcomes on δ cells and β cells. It would be useful to clarify this more in the text and to document the δ cell hypertrophy in the supplemental data.

---

## [Author Response]

In general, there was a lot of enthusiasm for this interesting work which will enhance our understanding of pancreatic islet innervation, an area that is currently not well understood. However, despite their enthusiasm, there are a number of issues that the reviewers agree need to be resolved before this work can be considered for publication. Foremost among these is that your study does not reveal the origin of the Elav-expressing cells whose development you have described. Although you show that neural crest is required for pancreatic islet innervation, your study does not provide evidence that these Elav cells are neural crest derived, and thus does not rule out the possibility that they have an endocrine origin. Thus, you will need to investigate both whether neural crest cells and whether endocrine cells from the pancreas contribute to this Elav-expressing population. Sox10 may not be expressed early enough to be definitive for this experiment, thus you may need to use other transgenic lines to look at earlier developmental stages. This analysis will also provide important information on the timing of association of these cells with the developing pancreas.

We thank the reviewers for this comment and have now employed two different methods to address the origin of the cells displaying *elavl3* promoteractivity. First lineage tracing with *Tg(sox10:CreERT2, myl7:GFP); Tg(ubb:loxP-CFP-loxP-nuc-mCherry)* zebrafish upon tamoxifen treatment from 16 to 24 hpf in combination with the *elavl3* reporter line was used to show that the *elavl3*+ migratory population of cells were once positive for *sox10* promoter activity (as seen in new Figure 3B and updated Figure 4). Additional lineage tracing experiments also indicated that neural crest derived cells were not positive for Insulin, Glucagon, or Somatostatin immunostaining (Figure 4B). Second, we conducted time-lapse imaging of *Tg(sox10:GAL4); Tg(UAS:GFP); Tg(sst2:RFP)* zebrafish and observed a similar subset of cells migrating away from the islet (Figure 3C). These cells were not positive for Insulin or Somatostatin. These results are now included in the new Figure 3. Although the *elavl3* promoter is active in a subpopulation of endocrine cells in early stages of development (Figure 2B), our data suggest that they do not contribute to the subset of cells migrating away from the islet.

There are a number of other issues that should also be addressed in your resubmission.1) We suggest changing the title to reflect the fact that you only investigate pancreatic islet innervation.

We have changed the title to “A new mode of pancreatic islet innervation revealed by live imaging in zebrafish”.

2) It would be useful to distinguish neurites derived from the vagus nerve from those of the Elav-expressing cells.

We thank the reviewers for this comment and agree that it would be useful to distinguish vagus nerve neurites from those of the detached *elavl3* expressing cells. We have attempted to conduct lineage tracing with photoconversion of *Tg(elavl3:GAL4-VP16); Tg(UAS:Dendra); Tg(sst2:RFP),* but have not been able to observe any photo-converted cells at the 80 hpf timepoint, likely due to the degradation of the photo-converted protein over time. Without an inducible CreER transgenic line that could be used to selectively mark this sub-cluster of neurons, we are currently unable to lineage trace with permanent expression of a reporter. We hope that the reviewers understand the technical difficulties related to this experiment. We have however now stained *Tg(elavl3:GAL4-VP16); Tg(UAS:EGFP-CAAX); Tg(sst2:RFP)* zebrafish with acetylated tubulin (Figure 2—figure supplement 1). Although this approach is not sufficient to quantify nerve density arising from the vagus versus the *elavl3* expressing detached neurons, it reveals that at least a subset of nerves are extending from neurons that appear to have detached from the islet. We have included a statement in the manuscript addressing this concern that without lineage tracing, it is hard to distinguish the nerves from the detached neurons versus those from the vagus.

3) It is difficult to see the detachment of the Elav-expressing cells in your movies, making it unclear whether the later cells arise from proliferation or cells that turn on Elav expression at a later time.

We have now adjusted the contrast to help the visualization of these cells.

4) The laser-directed, ntr-directed, and botulinic toxin-directed ablations have different outcomes on δ cells and β cells. It would be useful to clarify this more in the text and to document the δ cell hypertrophy in the supplemental data.

We have now discussed the differences in the outcomes of the various models. The quantification of δ cell hypertrophy in the *Tg(elavl3:GAL4-VP16); Tg(UAS:BoTxBLC-GFP)* model is now included in Figure 7—figure supplement 1.